# TANGO1 membrane helices create a lipid diffusion barrier at curved membranes

Ishier Raote[1]*, Andreas M Ernst[2], Felix Campelo[3], James E Rothman[2], Frederic Pincet[2,4]*, Vivek Malhotra[1,5,6]*

[1]Centre for Genomic Regulation, The Barcelona Institute of Science and Technology, Barcelona, Spain; [2]Department of Cell Biology, Yale School of Medicine, New Haven, United States; [3]ICFO-Institut de Ciencies Fotoniques, The Barcelona Institute of Science and Technology, Castelldefels, Spain; [4]Laboratoire de Physique de l'Ecole normale supérieure, ENS, Université PSL, CNRS, Sorbonne Université, Université de Paris, Paris, France; [5]Universitat Pompeu Fabra, Barcelona, Spain; [6]Institució Catalana de Recerca i Estudis Avançats, Barcelona, Spain

**Abstract** We have previously shown TANGO1 organises membranes at the interface of the endoplasmic reticulum (ER) and ERGIC/Golgi (Raote et al., 2018). TANGO1 corrals retrograde membranes at ER exit sites to create an export conduit. Here the retrograde membrane is, in itself, an anterograde carrier. This mode of forward transport necessitates a mechanism to prevent membrane mixing between ER and the retrograde membrane. TANGO1 has an unusual membrane helix organisation, composed of one membrane-spanning helix (TM) and another that penetrates the inner leaflet (IM). We have reconstituted these membrane helices in model membranes and shown that TM and IM together reduce the flow of lipids at a region of defined shape. We have also shown that the helices align TANGO1 around an ER exit site. We suggest this is a mechanism to prevent membrane mixing during TANGO1-mediated transfer of bulky secretory cargos from the ER to the ERGIC/Golgi via a tunnel.

*For correspondence:
ishier.raote@crg.eu (IR);
frederic.pincet@ens.fr (FP);
vivek.malhotra@crg.eu (VM)

## Introduction

Intracellular membrane trafficking is key for the homeostatic regulation of compositional gradients of proteins and lipids, required to maintain organelle identity along the secretory pathway (*von Blume and Hausser, 2019*; *Guo et al., 2014*; *Holthuis and Menon, 2014*; *van Meer and Lisman, 2002*; *van Meer et al., 2008*). These gradients are established by the formation and transport of highly curved transport intermediaries; a key question is whether (and how) membrane curvature plays a major role in lipid and protein sorting (*Campelo and Malhotra, 2012*). While it is clear that lipid sorting occurs during the biogenesis of transport intermediates, current evidence suggests that curvature-mediated lipid sorting is not based on curvature alone, but requires additional protein-lipid interactions (*Brügger et al., 2000*; *Callan-Jones et al., 2011*; *Gruenberg, 2003*; *Klemm et al., 2009*; *Roux et al., 2005*; *Sorre et al., 2009*; *Tian and Baumgart, 2009*).

Protein and lipid sorting during export from the endoplasmic reticulum (ER) utilises ER export machinery including the multisubunit COPII complex, which controls the formation of cargo-containing transport intermediates. In vitro preparations of COPII vesicles show a differential lipidomic profile from the ER, including an enrichment in lysolipids such as lysophosphatidylinositol (LPI) and a decrease in phosphatidylserine (*Melero et al., 2018*). COPII vesicles form at ER exit sites (ERES), which are semi-stable specialized subdomains of the ER with structures of a pleomorphic organization of diverse curvatures, including cup-like shapes and beaded tubes (*Bannykh et al., 1996*; *Hughes et al., 2009*). How ERES morphology is established and maintained, and how these

structures contribute to cargo export and protein/lipid sorting between the ER and subsequent secretory compartments, are still open questions.

A unique challenge during ERES-mediated lipid sorting arises during the export of bulky and complex cargoes such as procollagens from the ER. During procollagen export, the ER and the post-ER compartment (ERGIC/Golgi) are transiently connected, yet there is a controlled transfer of proteins and lipids between the two compartments. Procollagens in the ER fold and trimerize into rigid, rod-like elements that are considered too large for conventional COPII coated vesicle-mediated transport (*Burgeson et al., 1985*; *Gorur et al., 2017*; *Kadler, 2017*; *Kadler et al., 2007*; *McCaughey and Stephens, 2019*; *Omari et al., 2018*). Due to large cargo size, it has been hypothesized that large COPII-coated carriers mediate procollagen export from the ER to the Golgi. Outsize COPII carriers were reported in cells overexpressing KLHL12, which generates large procollagen-containing structures that co-label with Sec31A (*Jin et al., 2012*; *Yuan et al., 2017*; *Yuan et al., 2018*). A subsequent report suggested that these structures are destined for degradation and not *bona fide* transport carriers (*Omari et al., 2018*). Alternatively, it has been proposed that coordinated assembly of inner and outer COPII coats can mediate the biogenesis of vesicles ranging from small spheres to large tubular carriers, commensurate with the size of bulky cargoes (*Hutchings and Zanetti, 2019*; *Hutchings et al., 2018*; *Stagg et al., 2008*; *Zanetti et al., 2013*). However, ER-to-Golgi procollagen carriers appear pleomorphic and do not seem to contain any detectable COPII (*Mironov et al., 2003*).

More recent evidence suggests an entirely different mechanism is at play. Procollagen export from the ER requires the assembly of a functional machine centred on the transmembrane ERES-resident protein TANGO1 (*Ishikawa et al., 2016*; *Lekszas et al., 2020*; *Saito et al., 2009*; *Wilson et al., 2011*). As a master regulator of ERES assembly, TANGO1 acts as a filamentous linactant to recruit, constrain, and scaffold ERES machinery and post-ER (ERGIC) membranes for procollagen export (*Ma and Goldberg, 2016*; *Maeda et al., 2017*; *Nogueira et al., 2014*; *Raote et al., 2017*; *Raote et al., 2018*; *Raote et al., 2019*; *Saito and Maeda, 2019*; *Saito et al., 2009*; *Santos et al., 2015*).

We have proposed that a procollagen carrier is not formed by sculpting a vesicle from ER membrane as in the conventional model of COPII coated vesicle formation. Instead, TANGO1 utilises a retrograde membrane tether complex (NRZ complex) to tether post-ER membranes (ERGIC/Golgi), which fuse at the ERES to create an export route for procollagens. The membrane fusion creates a pore or tunnel between the ER and the ERGIC which is stabilised and constrains by TANGO1, as cargo is transferred for anterograde transport. Thus retrograde membrane (the ERGIC) is the anterograde carrier. During the process of cargo transfer, we propose the ERGIC and the ER are maintained as two distinct compartments, but how are their membrane lipids and proteins prevented from mixing completely?

Using super-resolution (STED) microscopy to visualise TANGO1, its interactors, and associated machinery we have described its assembly into a filament-like ring to organise the early secretory pathway (*Liu et al., 2017*; *Raote and Malhotra, 2019*; *Raote et al., 2017*; *Raote et al., 2018*; *Raote et al., 2019*; *Reynolds et al., 2019*). By assembling into a ring around an ERES, TANGO1 corrals COPII machinery and generates a semi-stable sub-domain across multiple compartments. Could TANGO1 also participate in preventing lipid mixing between the ERGIC and the ER? A ring of TANGO1 is ideally spatially organised to partially separate membrane inside the ring from the rest of the ER. Retrograde ERGIC membranes fuse within the ring, serving as an anterograde conduit for collagen. In other words, how are ERGIC membranes effectively partitioned from the rest of the ER, by a diffusion barrier created around the site of REGIC retrograde fusion at an ERES?

TANGO1 is a single-pass transmembrane protein with two adjacent hydrophobic helices; amino acid residues 1177–1197 of human TANGO1 are predicted to form a transmembrane helix, while residues 1145–1165 form a helix that penetrates the lumenal leaflet of the ER membrane (*Saito et al., 2009*). Together, these helices could mediate membrane partitioning. We have reconstituted the two hydrophobic helices in model membrane and show that they create a diffusion barrier for lipids, at a region of defined membrane curvature and shape. The diffusion barrier should occur at the base of the tube, paralleling the neck of a growing transport carrier/tunnel at an ERES.

Finally, using super resolution (STED) microscopy, we show that the functions mediated by the helices are incorporated into TANGO1 assembly into a ring around an ERES. We propose that such a barrier could sub-compartmentalise ER membrane, allowing retrograde ERGIC/Golgi membrane to be constrained during cargo transfer out of the ER, while maintaining the compositional identities of the individual compartments.

## Results and discussion

### TANGO1 membrane helices partition lipids

In order to investigate whether the membrane-associated helices of TANGO1 can contribute to the organisation of the ER-ERES boundary, we employed recombinant minimal transmembrane proteins that were reconstituted in model membranes. In order to mimic the marked changes in curvature that are encountered at this boundary, we employed micro-manipulation of giant unilamellar vesicles (GUV). Here, a tube with a diameter of 50–200 nm is pulled from the surface of the GUV, providing an ideal proxy to characterise membrane reorganisation and the role of membrane curvature and shape during transport intermediate biogenesis at the ER.

We purified the intramembrane (IM) and transmembrane (TM) helices, tagged with monomeric superfolder enhanced GFP (IM-TM_GFP). As an experimental control, we used the TM_GFP alone. Proteins were expressed and purified (*Figure 1A*) using EXPI293F cells, as before (*Ernst et al., 2018*). Proteo-GUVs were generated using an osmotic shock as described before (*Motta et al., 2015*), with a lipid mixture of POPC:DSPE-PEG-biot:DOPE-Atto647N; 95: 5: 0.1. Atto647N was used to visualize lipids in the GUV membrane.

To confirm that similar levels of protein were incorporated into GUVs in the two conditions tested, we compared the intensities of IM-TM and TM in the GUVs and found that they are similar under the same conditions of microscope objective, laser power and gain: $92 \pm 23$ a.u. for IM-TM and $106 \pm 38$ a.u. for TM respectively. The lipid intensities are also the same, $191 \pm 61$ a.u. and $178 \pm 37$ a.u. for IM-TM and TM respectively. All values are mean $\pm$ standard deviation.

These fluorescence intensities can be used to arrive at an approximation of the ratio of protein to lipids. Under our conditions, we estimate 1 IM-TM protein per 18,000 lipids, that is a typical inter-protein distance of 80 nm (see Materials and methods section for a more detailed description of this quantification).

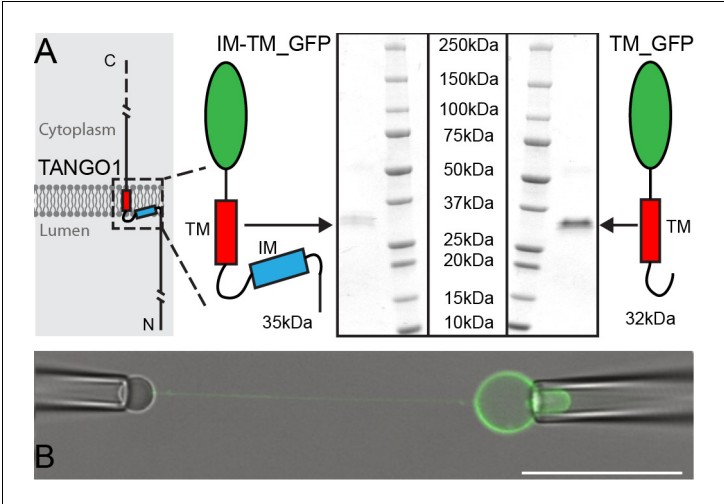

**Figure 1.** Micromanipulation of TANGO1 transmembrane-domain minimal constructs reconstituted into GUVs. (**A**) The two constructs used in this study. Schematics and Coomassie gels of the purified proteins with either both hydrophobic helices (Intramembrane IM, and transmembrane TM) tagged with monomeric superfolder EGFP (IM-TM_GFP left), or the transmembrane alone with the same EGFP tag (TM_GFP right). (**B**) The proteins (green) were reconstituted into GUVs and a second pipette with a biotin-tagged bead was used to pull a tube from the surface of the GUV. Scale bar 25 μm.

A micropipette was used to grab the GUV, and a streptavidin-coated silica bead was brought in contact with the GUV for a few seconds to form streptavidin-biotin bonds. Separation of the bead from the GUV led to a ~ 100 µm long membrane tube, thereby generating a site of saddle-like curvature at the junction of tube and the flat GUV membrane (*Figure 1B*). The aspiration pressure in the GUV micropipette was varied between 5 Pa and 100 Pa, which in our geometry corresponds to tube diameters between 200 nm and 50 nm. We did not observe any dependency in our experiments on the aspiration pressure; hence this is not discussed further.

To observe lipid diffusion across the region of saddle-like curvature between the tube and the GUV, we photobleached Atto647N in the tube membrane and quantified its recovery after photobleaching. Fluorescence recovery in the tube was due to the diffusion of labelled lipids from the GUV into the tube. This experiment was carried out in GUVs under three conditions: with no protein (no protein), with TM_GFP (TM), and with IM-TM_GFP (IM-TM) as depicted schematically in *Figure 2A*. Images of Atto647N in the tube were acquired before bleaching (prebleach), immediately after the photobleach (0 min) and six minutes later (6 min) (*Figure 2B*). The recovery of fluorescence was plotted six minutes after photobleaching (*Figure 2C*). This single time point was chosen such that fluorescence recovery in the tube had plateaued (see Materials and methods section for

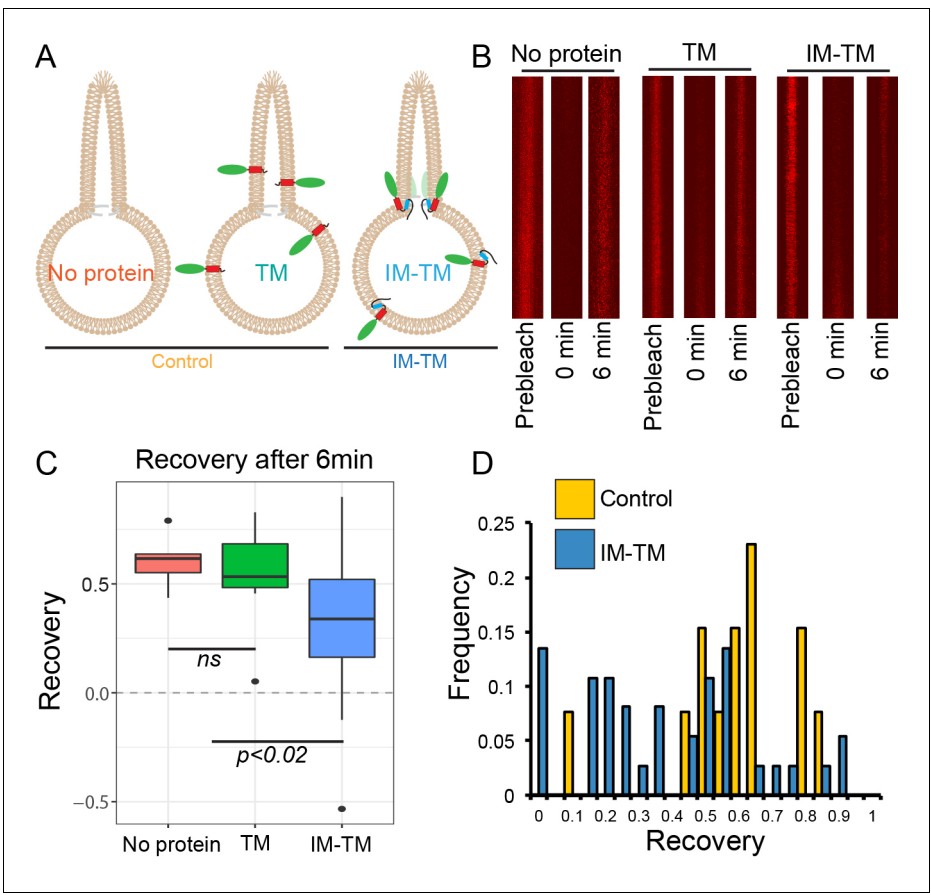

**Figure 2.** TANGO1 helices are a barrier to diffusion of lipids. (**A**) Schematic of the three conditions (no protein, TM, IM-TM) with protein reconstitution into a GUV showing a tube pulled from the GUV; (**B**) Atto647N-labelled lipids visualised in the pulled tube in all three conditions, before bleaching (prebleach), then immediately after bleaching (0 min) and six minutes later (6 min). (**C**) Box plot quantification of the fluorescence recovery after 6 min; (**D**) Frequency histogram showing the extent of fluorescence recovery in tubes pulled from GUVs with no protein or TM pooled (yellow) or GUVs with TM_IM (blue). Most trials with low recovery rates were those where tubes were pulled from GUVs with IM-TM (blue line vs green and red lines). ns – not significant or p<0.02 by ANOVA.

The online version of this article includes the following figure supplement(s) for figure 2:

**Figure supplement 1.** IM-TM diffuses freely.

details). Fluorescence recovered to 60 ± 6% (mean ±s.d.) of the prebleach intensity in tubes from GUVs with no protein. Similarly, in tubes pulled from GUVs with the transmembrane helix alone (TM), fluorescence recovered to 56 ± 12%. However, the two helices together (IM-TM), considerably inhibited fluorescence recovery. Now, the recovery was only 35 ± 5% (*Figure 2C*), showing that TANGO1 membrane helices together inhibited the exchange of lipids between the tube and the rest of the GUV.

Our observations will probably under-represent the magnitude of the effect; many trials showed no effect due to several confounds including, that proteins were disrupted by a movement of the membranes, drift of the pipette, slight variation of pressure in the pipette, convective flow in the buffer, etc. This is made clear in an alternate representation of the data, shown in a frequency histogram of all the trials together (*Figure 2D*). The two treatments, no protein and TM, are grouped as control, and compared with the trials with 'IM-TM'. Results from the trials with IM-TM (blue bars) appear to divide into two peaks, one with a lower recovery rate and a second peak that looks like control trials. Most trials with low recovery rates were those observed in tubes pulled from GUVs with IM-TM. There were almost no trials in control conditions (either no protein, or TM alone) with slowed recovery. We expect that those trials that showed no effect might have been affected by one or more of the confounds described above.

To make sure the low recovery in the tube is not due to a global reduced fluidity of the membrane of GUV with IM-TM, we performed fluorescence recovery after photobleaching (FRAP) experiments on the EGFP-labeled protein in the GUV (*Figure 2—figure supplement 1A*). We found a diffusion coefficient (0.7 ± 0.3 $\mu m^2/s$), consistent with a protein that is monomeric or in the form of small oligomers (*Figure 2—figure supplement 1B*). These data show that the membrane (protein and lipid) is fully fluid without hindered diffusion, confirming that the low recovery in the tube is due to reduced crossing at the junction between the GUV and the tube.

In treatments where lipid flux between the GUV and the tube is slowed or blocked, we expect a similar or even more pronounced behavior for transmembrane proteins. To test this hypothesis, we attempted to use the same photobleaching and recovery assay to estimate the flux of GFP-associated fluorescence between the GUV and the tube. Unfortunately, we were unable to accurately quantify the GFP-associated fluorescence in the tube, as the protein levels in the tube are too low to be clearly distinguished from the background and are rapidly bleached during imaging.

Such a diffusion barrier could most likely form if the membrane helices are concentrated and retained at the base of the pulled tube. We were unable to test this prediction and visualise an immobile pool of protein at the base of the tube under our experimental conditions, as the diameter of the tube is 50–200 nm and a stable pool of GFP fluorescence is indistinguishable from the fluorescence in the rest of the GUV, even after photobleaching the bulk of the GUV and attempting to visualise GFP signal retained at the base of the tube. Visualising such a cluster of a few molecules might be possible using total internal reflection (TIRF) microscopy. Our procedure does not permit this type of single-molecule measurements; we used a confocal microscope with a dry objective resulting in a relatively high background from the surrounding membrane. Additionally the dyes used are neither bright nor photostable enough for such analyses.

These results showed that the two TANGO1 helices were sufficient to partially restrict lipid exchange between the tube and the rest of the GUV. What role do the two helices play in how TANGO1 interacts with the ER membrane?

## The intramembrane helix is not required to target TANGO1 to ERES

Proteins that insert into the lipid bilayer can bend membranes by changing the local spontaneous curvature of the monolayer or, by being retained at a location, they can stabilise a specific shape of membrane (*Kozlov et al., 2014*). We envisage that TANGO1 helices together are retained at, and/or stabilise, the saddle-shape membrane at the junction of the tube and the GUV, as at the junction of the ERES and a transport intermediate. Thus, we propose that the pair of helices act as a membrane shape-sensing module.

What could TANGO1 use this shape sensor for, in vivo? A first possibility we could test is that it promotes the correct localisation of TANGO1 in the ER. The ability to detect defined membrane shapes and curvatures could be utilised by TANGO1 to localise to the site of defined shape at an ERES, particularly the junction of the ER and an export intermediate. We have previously shown that TANGO1 is recruited to ERES via an interaction between its C-terminal proline rich domain (PRD)

and ERES proteins Sec23A and Sec16A. Perhaps the membrane shape-sensitivity of TANGO1 would play an additional role in its localisation as well. To test the role of the putative shape sensor in targeting TANGO1 to an ERES, we deleted the IM helix (amino acids 1145–1165, TANGO1ΔIM) or the TM (amino acids 1177–1197, TANGO1ΔTM) from full length TANGO1 (schematic representation is shown in *Figure 3A*) and assayed for the ability of the resulting proteins to localise to ERES. We transfected these constructs in HeLa cells from which endogenous TANGO1 has been deleted using CRISPR/Cas9 methodology (named 2H5 cells), as described before (*Raote et al., 2018*; *Santos et al., 2015*), and observed their pattern of expression. Deletion of the IM helix had no discernible effect on the ability of TANGO1 (*Figure 3B*, green) to localise to ERES as visualised by its localisation to puncta of the ERES marker Sec16A (*Figure 3B*, red).

Deletion of the TM helix had a more substantive effect and resulted in a loss of TANGO1 localisation to ERES (*Figure 3A,B*, quantified in 3C). We have previously shown that when the TM in TANGO1 is deleted, the IM changes its topology and becomes a transmembrane. Deleting both hydrophobic helices leads to a soluble protein in the ER lumen (*Saito et al., 2009*).

As a control, we confirmed that deleting either helix had no effect on targeting TANGO1 to accumulations of procollagen VII in the ER (*Figure 3—figure supplement 1*). Both TANGO1 and TANGO1ΔIM were recruited as small, discrete puncta, which were apposed to collagen accumulations in the ER. The TANGO1ΔTM entirely overlapped with procollagen VII accumulations.

Together, these data reveal that membrane curvature/shape sensitivity in TANGO1 membrane helices is not the dominant ERES-targeting device, instead the PRD is more important to target TANGO1 to its cellular location at an ERES. If the membrane shape sensor is not to target TANGO1 to ERES or collagen, what role does it play?

## TANGO1 membrane helices confer direction to TANGO1 molecules in a ring

TANGO1 and procollagen VII were co-expressed in HeLa cells from which TANGO1 has been knocked out using the CRISPR/Cas9 system (2H5 cells). As before, transfected cells were fixed and processed for super-resolution (STED) imaging and TANGO1 was imaged at accumulations of procollagen in the ER. TANGO1 can be visualised assembled into rings at ERES (*Raote et al., 2017*; *Raote et al., 2018*). TANGO1-HA was visualised with two different antibodies raised against two distinct epitopes in the full-length protein. One epitope is in the ER lumen (corresponding to amino acids 472–525 of TANGO1), while the other is a cytoplasmic C-terminal HA-epitope (for schematic, *Figure 4A*). Under these conditions there was a clear separation of the signal from antibodies directed against the two different epitopes and we could identify two distinct configurations of the ring of TANGO1. Either the C terminus coalesced into a single spot within a ring formed by the lumenal antibody, or both could be visualised as concentric/colocalized rings (*Figure 4A* red vs. green respectively).

We propose that these distinct sets of images represented discrete configurations of TANGO1 during the formation of a carrier at an ERES. Initially, the C termini of TANGO1 lie within the ring, overlying the site of the formation of a carrier. As the carrier grows, the C termini are gradually pushed apart (for a schematic, *Figure 4D*).

We used STED microscopy to monitor the organisation of TANGO1 into rings by imaging samples stained for both the TANGO1 and HA epitopes. Qualitatively, we observed that abrogating the intramembrane helix reduced the frequency with which we observed rings, but this observation could not be quantified given the subjective nature of the selection of fields to image. Interestingly, in several of the rings that did form, the C termini (HA epitope) and lumenal portions of TANGO1 were now misaligned (*Figure 4B*). Now, the C termini were often randomly oriented with respect to the ring formed by the lumenal epitope.

Alternatively, we mutated two hydrophobic residues in the IM helix to charged residues (phenylalanine 1152 and 1154 to arginine – TANGO1-FMF-RMR), such that it would no longer form a hydrophobic helix (*Figure 4—figure supplement 1*). Here too, many rings of TANGO1 showed altered alignment of the two epitopes.

In such a ring, a nascent bud will be encircled by the ring and serve to recruit and stabilise a membrane-shape-sensitive TANGO1 to the site. In other words, ring formation could create a feedback loop through orientation/shape sensing to further reinforce rings.

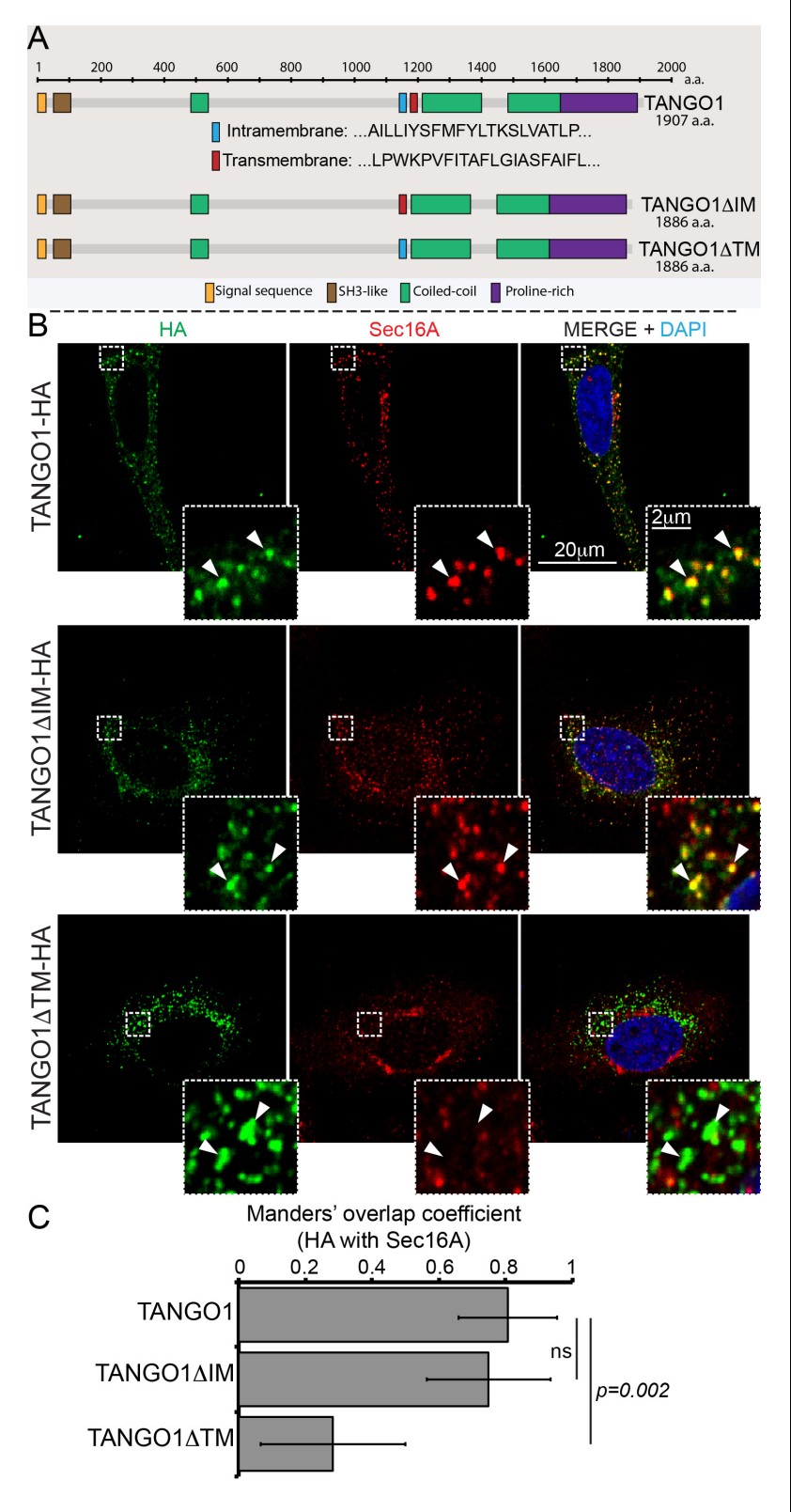

**Figure 3.** Deletion of the intramembrane helix has no effect on TANGO1 recruitment to ERES. (**A**) Schematic of TANGO1 constructs used in this study, showing the sequence of the transmembrane and intramembrane helices. (**B**) The three HA-epitope tagged constructs were expressed in 2H5 cells, which were then imaged for HA and

*Figure 3 continued on next page*

*Figure 3 continued*

Sec16A. Scale bar: 20µm, Inset: 2µm. (C) Plot of the Manders' colocalization coefficients of the extent of overlap of HA (TANGO1 constructs) with Sec16A.

The online version of this article includes the following figure supplement(s) for figure 3:

**Figure supplement 1.** Deletion of the intramembrane has no effect on TANGO1 recruitment to collagen.

Regulating lipid sorting and trafficking is key in organelle homoeostasis, to control all membrane traffic, and spatiotemporal modulation of membrane curvature is essential in this control. We showed that the transmembrane organisation of TANGO1 is sufficient to act as a diffusion barrier for membrane lipids. In doing so, our data suggest that TANGO1 is concentrated at the neck of a budding membrane and can act as a fence, constraining the flow of lipids.

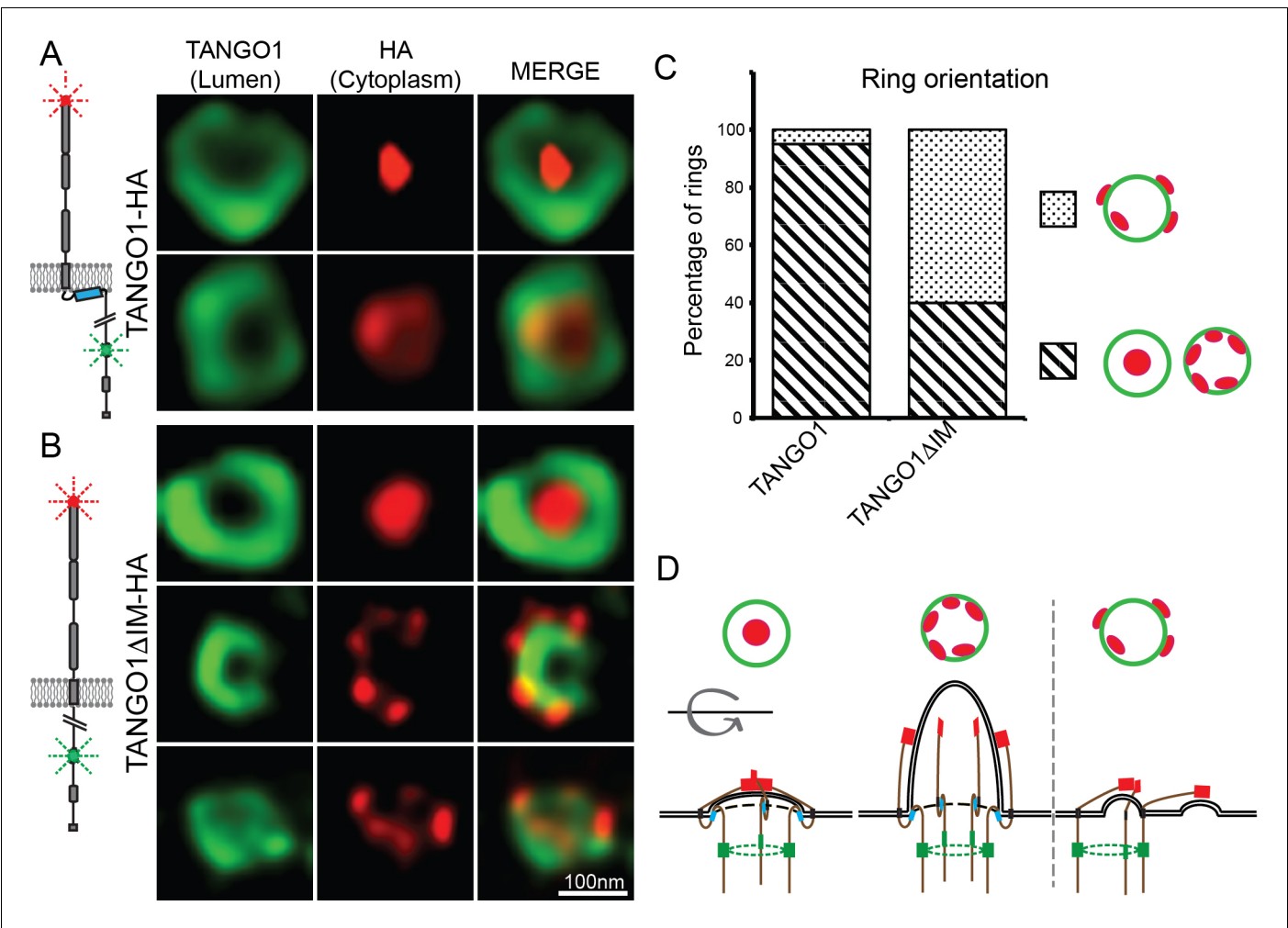

**Figure 4.** TANGO1 intramembrane helix orients a ring of TANGO1 molecules. Two different epitopes on TANGO1 are visualised concomitantly. The ER-lumenal epitope in green, the c-terminal HA epitope in red (A) TANGO1, (B) TANGO1ΔIM. (C) Quantification of the number of rings observed in each configuration. Regions with diagonal bars refer to rings with HA signal (red) contained within the lumenal signal (green), while the dotted region refers to rings with at least some HA signal (red) contained outside, but apposed to, the lumenal signal (green). (D) Schematic model of the different configurations of TANGO1 molecules in a ring.

The online version of this article includes the following figure supplement(s) for figure 4:

**Figure supplement 1.** Targeted disruption of the intramembrane helix, replacing two phenylalanines with arginines (TANGO1-FMF-RMR), mimics the effect of deleting the helix entirely.

We have proposed that cargo transport through the secretory pathway can be mediated by transient tunnels or pores between successive compartments (*Raote and Malhotra, 2019*). We have shown that retrograde fusion of ERGIC membrane to the ER could create a tunnel between the two compartments for procollagen export (*Nogueira et al., 2014*). How are two compartments, joined by a transient tunnel, kept biochemically distinct? How do COP coats, conventional cargo receptors, cargo proteins, and TANGO1 function as gatekeepers between the ER and the ERGIC? Lipids such as PI4P are required at the ERGIC-ERES interface to assist in the assembly of COPII-coated vesicles at ERES and hence traffic from the ER (*Blumental-Perry et al., 2006*; *Farhan et al., 2008*). Yet, if lipids like PI4P are largely restricted to the ERGIC/Golgi, how are they prevented from flowing from the ERGIC/Golgi into the bulk ER membrane?

The lipid flux through the partial barrier between the GUV and the tube is proportional to the effective perimeter of the saddle. From the recovery rates observed in *Figure 2*, then 50–75% of the saddle is blocked by the membrane helices.

To what extent might these numbers recapitulate events at an ERES? There are ~172,000 molecules of TANGO1 and 250–500 exit sites per cell or 350–700 TANGO1 molecules per exit site, surrounding transport intermediates (*Hammond and Glick, 2000*; *Itzhak et al., 2016*). Further reconstitution and modeling will be informative in delineating how this number of proteins generates the perimeter block. In vivo, there will be further effects from other associated ERES proteins including the TANGO1 family protein cTAGE5 (*Saito et al., 2011*).

Our data presented here suggest that by assembling into a ring at an ERES, TANGO1 can act to reduce lipid flow at the boundary between the two compartments, delaying lipid and most likely transmembrane protein mixing between them.

We envisage that the recruitment of TANGO1 to membrane regions of specific shape and curvature will make for greater efficiency in generating a bud at an ERES. Neighbouring TANGO1 molecules in the ER membrane will align along curved membrane - found at an ERES. This process, whereby a TANGO1 fence limiting the coat polymerisation zone, could be homologous to the roles played by proteins which contribute to the initiation of clathrin-mediated endocytosis FCHO1/2 and EPS15/EPSR (*Avinoam et al., 2015*; *Ma et al., 2016*). Hundreds of TANGO1 molecules, at each site, will have an additive or even synergistic effect in stabilising the bud. As the site of budding is defined, the increased membrane curvature will promote the recruitment and localisation of curvature-sensitive components including Sar1 and COPII coat proteins. These proteins will serve to reinforce the curvature stabilisation, in effect setting up a positive feedback loop, where exit site machinery and TANGO1 mutually recruit and constrain each other through their membrane sculpting capabilities. Aligned TANGO1 molecules will be constrained into a configuration in which it is more likely for them to participate in lateral interactions with each other, promoting their assembly into a filament, further reinforcing the organisation of TANGO1 and its interactors at the site.

These reinforcing activities are evident from our data, showing that only half of the rings of TANGO1ΔIM show detectable signal outside the ring from the C-terminal epitope (*Figure 4*). Each ring is likely comprised of hundreds, of TANGO1ΔIM molecules. For signal from the two different epitopes to appear normal, most molecules must be constrained in the correct orientation, probably by the combined effects of other proteins at the site including COPII components and other TANGO1 family proteins.

It is possible that the recruitment/stabilisation of TANGO1 at a region of defined saddle-like shape will contribute partially to the retention of TANGO1 in the ER instead of accompanying cargo to the next secretory compartment.

Do other proteins exhibit a similar pair of helices, and could this combination of helices represent a general mechanism to sense or stabilise membrane curvature and modulate lipid diffusion at sites of contact between two compartments? Proteins related to TANGO1 also have two adjacent membrane helices; an alternatively spliced short variant of TANGO1 (TANGO1-short) lacks lumenal domains, but still has two helices though its IM helix has a different sequence. The TANGO1-like protein TALI (or MIA2) also has two membrane helices and it too is a single-pass transmembrane protein. Interestingly, a recent study described another single-pass transmembrane protein TMEM131 with two adjacent predicted membrane helices, which functions at the interface of two compartments. Again one of these helices must span the membrane, while the other is likely inserted into one leaflet of a compartment membrane (*Zhang et al., 2020*).

In sum, TANGO1 through its various parts select and bind cargoes like the procollagens in the lumen, and interacts with coats and other cytoplasmic components that tether and transiently fuse ERES to ERGIC or the cis-Golgi (in cells that lack an ERGIC). The transmembrane helices in TANGO1 assist in its alignment into a ring to prevent membrane protein and lipid mixing during the transient coupling of these two secretory compartments. This transient pore or tunnel thus allows cargo movement while retaining compartmental specificity.

## Materials and methods

### Cell culture and transfection

HeLa cells were grown at 37°C with 5% CO2 in complete DMEM with 10% FBS unless otherwise stated. Plasmids were transfected in HeLa cells with TransIT-HeLa MONSTER (Mirus Bio LLC) or Lipofectamine 3000 Transfection Reagent (Thermo Fisher Scientific) according to the manufacturer's protocols. All cells in culture were tested every month to confirm they were clear of contamination by mycoplasma.

### Molecular biology

All molecular cloning, of constructs with TANGO1, was carried out using MAX Efficiency Stbl2 Competent Cells (Thermo Fisher Scientific), following manufacturer's instructions.

### Antibodies

The following antibodies were used procollagen VII (rabbit anti–human [Abcam]; mouse anti–human [Sigma-Aldrich]), Sec31A (mouse anti–human; BD), TANGO1 (rabbit anti–human; Sigma-Aldrich), sec16A (rabbit anti-human; Sigma-Aldrich), calreticulin (goat anti–human; Enzo Life Sciences), HA (mouse; BioLegend), TGN46 (sheep polyclonal, Bio-Rad), HA (mouse monoclonal, BioLegend; rat monoclonal BioLegend). Mounting media used in confocal and STED microscopy were either Vectashield (Vector Laboratories) or ProLong (Thermo Fisher Scientific, Waltham, Massachusetts).

### Protein expression

Proteins were purified as previously described (*Ernst et al., 2018*). In brief, Plasmid encoding for FLAG-tagged TM or IM-TM were transfected into EXPI293F cells according to the manufacturer (Thermo Fisher), and incubated at 37°C, 8% $CO_2$ for 48 hr, pelleted, and resuspended in 50 mM HEPES/KOH pH 7.3, 175 mM NaCl, 5 mM EDTA, 1 mM PMSF, 1 mM TCEP, protease inhibitor cocktail, and 8% (v/v) TX-100. After a brief microtip sonication, the lysate was placed rotating at 4°C for 3 hr. After centrifugation of the cell debris, the lysate was incubated with FLAG-affinity resin for 2 hr at 4°C. The resin was added to a column, settled, and washed three times with 50 mM HEPES/KOH pH 7.3, 175 mM NaCl, 5 mM EDTA, 50 mM n-Octyl-β-D-glucopyranoside (OG), and proteinase inhibitor (Roche). Finally, the proteins were eluted by adding 125 ng/ml FLAG peptide to the OG-containing buffer (Sigma-Aldrich) and incubating the beads for 30 min per round of elution. Eluates were analysed on 4–20% Bis-Tris gradient gels, stained with Coomassie, and analysed on a LI-COR Odyssey infrared scanner.

### Giant unilamellar vesicle (GUV) formation

The GUVs contained 1-palmitoyl-2-oleoyl-glycero-3-phosphocholine (POPC, Avanti Polar Lipids, product #850457C), 1,2-distearoyl-sn-glycero-3-phosphoethanolamine-N-[biotinyl(polyethylene glycol)−2000] (DSPE-PEG2000-Biotin, Avanti Polar Lipids, product # 880129C) and 1,2-Dioleoyl-sn-glycero-3-phosphoethanolamine were pro-Atto-647N (ATTO-647N-DOPE, Atto-tec) at a ratio 94.9:5:0.1 mol/mol. The first step was to prepare proteo-liposomes. 0.8 μmol of the lipid mixture (POPC:DSPE-PEG-biot:DOPE-Atto647N; 95: 5: 0.1) was deposited in a test tube and dried by nitrogen flow followed by desiccation for 30 min. 50 μl of either the TM or the IM-TM peptide were added to the tube and slowly vortexed at room temperature for 15 min to resuspend the lipids. 150 μl of 10 mM HEPES pH 7.4, 100 mM KCl buffer (190 mOsm) was then added to the tube while vortexing to ensure homogeneous mixing. This buffer addition formed proteo-liposomes by diluting OG below its micellar critical concentration (initial OG concentration: 50 mM, final concentration: 12.5 mM, CMC ~ 23 mM). The solution was then injected in a dialysis cassette, which was subsequently placed

in a large container with 4 l of 10 mM HEPES pH 7.4, 100 mM KCl buffer (190 mOsm). The container was placed in a cold room and the buffer stirred overnight. This dialysis ensured optimum removal of OG from the proteo-liposomes, which were subsequently transferred in a 500 μl microcentrifuge tube.

Guvs were produced using the osmotic shock method (*Motta et al., 2015*). In brief, a 2 μl drop of the proteo-liposomes was deposited on a MatTek dish (MatTek Corporation) and left to dry at room temperature. It was then rehydrated with a 2 μl water drop. The osmotic difference between the inside of the proteo-liposomes and the water drop immediately popped them and they resealed in a larger structure. The drying/rehydration cycle was repeated three more times to produce larger and more unilamellar proteo-GUVs (hereinafter called GUVs). The water drop was 6 μl for the third and fourth rehydration. After the last rehydration, water drops were deposited at the edge of the dish that was subsequently closed. This prevented any further evaporation while the GUVs are left to grow for an additional 30 min (or more). Afterwards, the water drops were removed and the MatTek dish was filled with the dialysis buffer diluted 2.5 times to slightly deflate the proteo-GUVs.

## Quantification of protein to lipid ratio in the GUVs

We plotted fluorescence intensity along a straight line across the GUV and measured the intensities of the two peaks corresponding to the membrane. We compared the intensities of lipids and GFP from IM-TM and TM in the GUVs under the same conditions of microscope objective, laser power and gain. Measurements were carried out on eight different GUVs from four independent experiments for IM-TM and three different GUVs from two independent experiments for TM. The protein intensity can provide a very approximate value of the concentration using a separate calibration with Atto488 dyes. We measured that Atto488 dyes are twice as bright as Alexa488 (*Figure 5A*) which itself is twice as bright as EGFP (quantum yield and brightness). In addition, increasing the gain of image by 100 leads to approximately a twofold increase in intensity. We made GUVs with various concentrations of Atto488 (from 0.05% to 2% lipids) and observed them with 5% laser power and a gain of 500. The result is a linear variation with Atto488 concentration (*Figure 5B*).

Using these calibration data and assuming a linear increase of the intensity with laser power, the value of 92 for the intensity of IM-TM with 20% laser power and 1250 gain leads to one protein per 18,000 lipids. It corresponds to an average of one protein per 6300 nm$^2$, that is a typical inter-protein distance of 80 nm. With our experimental conditions and parameters, this density corresponds to one protein per pixel at the pole of the GUVs, and 10 proteins per pixel at the equator of the GUV (because of the vertical projection). This rough estimate seems realistic with regard to the low intensity observed in our GUV experiments (see for instance the FRAP experiments described in the *Figure 2—figure supplement 1*).

Calculation of the protein density:

$$\left( \frac{92 * 2 * 2 * 5 / (20*)^{2.5} \left( \frac{1250-500}{100} \right)}{10768} \right) = 0.00056\%$$

Note that we needed to go through Atto488 intermediate because this is a bright dye, commercially available on lipids.

## Tube formation, bleaching and recovery

MatTek dishes were modified to have two diametrically opposed openings in the wall so that quasi-horizontal micropipettes can move downward and reach the bottom coverslip. After GUV formation, the MatTek dish was placed on a Leica SP8 confocal microscope and ~10,000 streptavidin coated silica beads (2 μm, Bangs Laboratories) were added. A first micropipette (inner diameter ~4 μm) was used to grab a proteo-GUV. A second one (inner diameter 1–2 μm) grabbed a bead. The aspiration in both micropipettes was controlled by a standard hydrostatic pressure system (relative accuracy: 1 Pa, absolute accuracy: 5 Pa, aspiration range: 5 Pa to 3,000 Pa). The GUV were then placed facing each other ~50 μm above the coverslip. The aspiration in the GUV was reduced to 5 Pa to lower the surface tension. Then, the bead was brought in contact with the GUV for a few seconds to ensure the formation of several streptavidin-biotin bonds. Upon separation of the bead from the GUV, a ~ 100 μm long tube of membrane was pulled, setting two connecting regions of very different curvature: the high curvature tube and the flat GUV membrane. The aspiration in the GUV micropipette

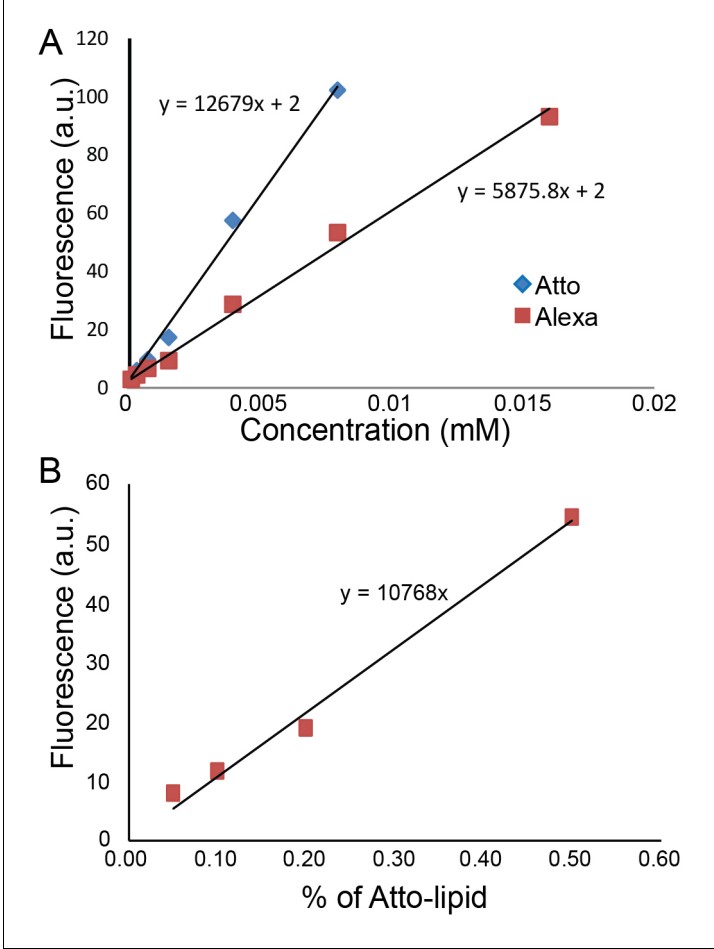

**Figure 5.** Quantification of protein concentration. (**A**) Alexa488 and Atto488 intensities in solution at different concentrations. (**B**) Atto488 lipid intensity increases linearly with concentration.

was increased to the desired value and the system was left alone to relax for a couple of minutes. We varied the aspiration between 5 Pa and 100 Pa, which in our geometry corresponds to tube diameters between 200 nm and 50 nm. We did not observe any dependency of the fluorescence recovery on the aspiration; hence this is not discussed in the results section.

A first picture of the tube was acquired to obtain the initial tube fluorescence intensity ($i_0$). Then, fluorescent lipids in the tube alone (not the GUV) were almost completely bleached by illumination with a 647 He-Ne laser at maximum power for a few seconds. A second picture was immediately acquired and the tube fluorescence intensity after bleaching was measured ($i_b$). Finally, a third picture was taken after 6 min, which provided the final tube fluorescence intensity ($i_6$). The recovery was established as ($i_6 - i_b$)/($i_0 - i_b$). We chose not to continuously monitor the fluorescence to avoid bleaching the fluorescent lipids as they migrate from the GUV to the tube.

We used 6 min because we wanted to be close to the fluorescence recovery plateau. We also wished to avoid spontaneous bleaching to have as accurate a measurement as possible and therefore acquired only one data point.

Diffusion time varies with the square of the typical dimension. Here we are working with 30 µm tubes, hence it takes about 10 times longer to reach the recovery plateau compared to a 10 µm experiment (size of a cell). Thus a recovery time of 30 s for 10 µm would take 5 min here.

Being more quantitative requires a better modeling of the geometry. Namely, the system of a tube blocked by a bead is equivalent to two mirror images (***Figure 6A***). With boundary conditions at t = 0 (***Figure 6B***):

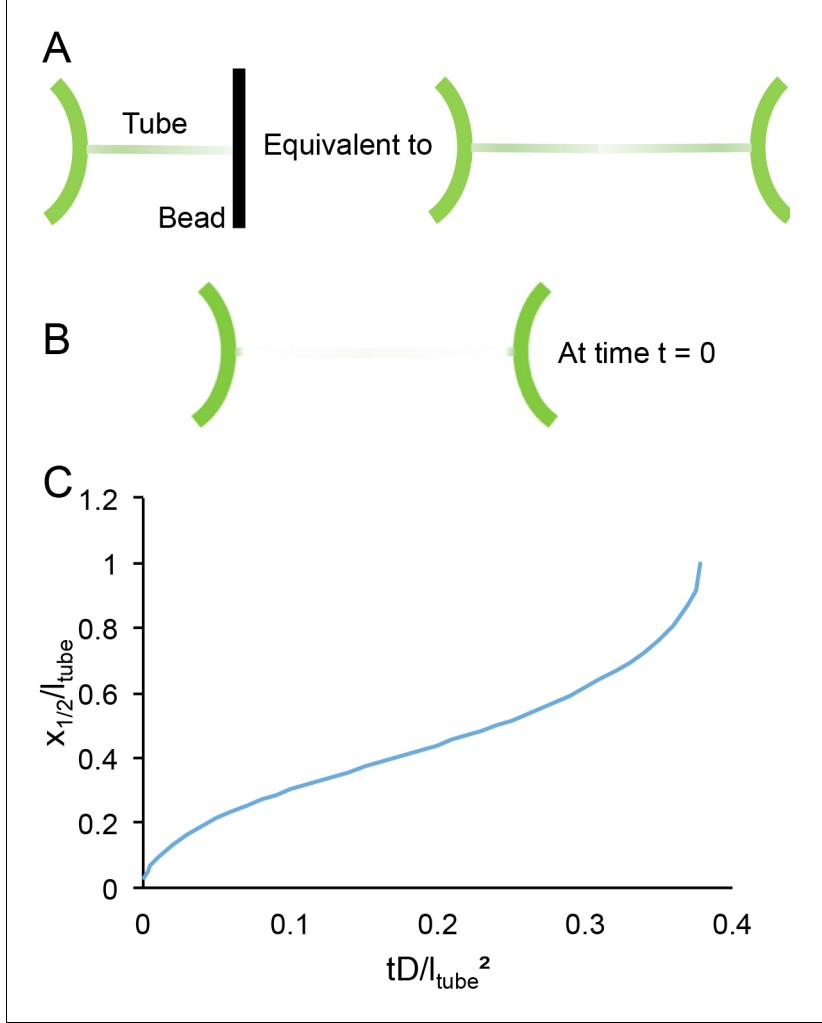

**Figure 6.** Time scales of lipid diffusion in the GUV-tube system. (**A**) Schematic of the system. (**B**) Schematic of system under boundary conditions at time t = 0. (**C**) Kinetics of recovery.

The 1D diffusion equation, $\frac{\partial \rho}{\partial t} - D\frac{\partial^2 \rho}{\partial x^2}\rho = 0$, can then be resolved numerically. A plot of the position of the mid recovery $x_{1/2}$ is a good representation of the kinetics (*Figure 6C*).

$x_{1/2}$ reaches the end of the tube at $t_{1/2} \sim 0.38\ l_{tube}^2/D$. The typical diffusion coefficient of a lipid in free membrane is ~5 $\mu m^2$/s for which $t_{1/2}$ is slightly more than 1 min. Even if the diffusion coefficient of the lipid is reduced five times (unlikely), 6 min would be sufficient to be close to the recovery plateau. Hence, the choice of 6 min is a good compromise between an optimum recovery and a minimal experimental risk (drift of the pipettes, rupture of the GUV, etc.).

FRAP-based quantification of protein diffusion in the GUV membrane:

Experiments were carried out using the protocol we previously described (*Pincet et al., 2016*). In brief, a GUV is micromanipulated and positioned slightly above the coverslip so that lower pole is positioned in the focal plane of the confocal microscope. A disk-shaped region of interest (ROI) with a known diameter *d* is chosen at the pole of the GUV. A couple of 'prebleach' images are acquired with the laser power adjusted at the minimum value providing an above-noise fluorescence signal from the ROI with minimal intrinsic bleaching. The EGFP signal from the GUV is dim, attesting the low protein concentration. This forced us to use high laser power, 20% of the maximum value and maximum gain. One image with 100% power was sufficient to completely bleach the fluorescence from the ROI. Then, recovery was recorded, typically at two images per second, at 20% of the laser power. The recovery phase was monitored over ~15–20 s. The raw data are very noisy (blue data points in *Figure 2—figure supplement 1*) but still display significant intrinsic bleaching, showing the

difficulty to adjust the laser power and obtain quantifiable measurements. To resolve this issue, we performed the rolling average over three successive images (orange data points in *Figure 2—figure supplement 1*). The data were normalized to one before bleach and 0 right after bleach. The resulting curve was fitted using the standard recovery behavior for diffusive species:

$$I(t) = I_\infty \, e^{-\frac{2\tau}{t}} + \left( J_0\left(\frac{2\tau}{t}\right) + J_1\left(\frac{2\tau}{t}\right) \right)$$

Where $J_0$ and $J_1$ are the modified Bessel functions of order 0 and 1, $I_\infty$ is the fluorescence at $t=\infty$ and $\tau$ is the characteristic diffusion time. The details of the calculation leading to *Equation (1)* are provided by *Soumpasis, 1983*. Note that $I_\infty$ is expected to be slightly below one when there is no intrinsic bleaching because a few percent of the total GUV area are bleached. Because of the spontaneous bleaching, the fit was performed over the first ~5 s of the recovery phase. The diffusion coefficient is equal to $d^2/(16\tau)$.

## Immunofluorescence microscopy

Cells grown on coverslips were fixed with cold methanol for 8 min at $-20°C$ or 4% formaldehyde (Ted Pella, Inc) for 15 min at room temperature. Cells fixed with formaldehyde were permeabilised with 0.1% Triton in PBS and then incubated with blocking reagent (Roche) or 2–5% BSA for 30 min at room temperature. Primary antibodies were diluted in blocking reagent or 2% BSA and incubated overnight at 4°C or at 37°C for 1 hr. Secondary antibodies conjugated with Alexa Fluor 594, 488, or 647 were diluted in blocking reagent and incubated for 1 hr at room temperature.

Confocal images were taken with a TCS SP5 (63×, 1.4–0.6 NA, oil, HCX PL APO), TCS SP8 (63×, 1.4 NA, oil, HC PL APO CS2), all from Leica Microsystems, using Leica acquisition software. Lasers and spectral detection bands were chosen for the optimal imaging of Alexa Fluor 488, 594, and 647 signals. Two-channel colocalization analysis was performed using ImageJ (National Institutes of Health).

## STED

STED images were taken on a TCS SP8 STED 3 × microscope (Leica Microsystems) on a DMI8 stand using a 100 × 1.4 NA oil HCS2 PL APO objective and a pulsed supercontinuum light source (white light laser). Images were acquired and deconvolved exactly as described before (*Raote et al., 2017*; *Raote et al., 2018*). In all images acquired and presented here, a goat-anti mouse secondary antibody, coupled to Abberior 635 was used to visualise anti-HA antibody localisation, while secondary antibody coupled to Alexa 594 was used to visualise the antibody targeted against the lumenal epitope of TANGO1. Axial resolution were found to be ~60 nm for the Abberior 635 channel and 50 nm for the Alexa Fluor 594 channel as described before (*Raote et al., 2017*).

## Acknowledgements

We thank Ivan Lopez Montero for discussion, Jose Wojnacki for discussions and assistance with data visualisation, Isabelle Motta for acquisition of the Atto 488/Alexa 488 calibration curves, Hong Zheng for assistance with protein purification. IR thanks Robert Ernst for discussions. We thank the Advanced Light Microscopy Unit at the CRG. V Malhotra is an Institució Catalana de Recerca i Estudis Avançats professor at the Centre for Genomic Regulation. FC acknowledges financial support from the Spanish Government ('Severo Ochoa' program for Centres of Excellence in R and D, SEV-2015–0522, and RYC-2017–22227), Fundació Privada Cellex, Fundació Privada Mir-Puig, and from the Generalitat de Catalunya through the CERCA program. The GUV experiments were performed at Yale with the support of NIH R35 GM118084 (JER). This work reflects only the authors' views, and the EU Community is not liable for any use that may be made of the information contained therein.

## Additional information

### Competing interests

Vivek Malhotra: Senior editor, *eLife*. Felix Campelo: Reviewing editor, *eLife*. The other authors declare that no competing interests exist.

### Funding

| Funder | Grant reference number | Author |
|---|---|---|
| Ministerio de Economía y Competitividad | SEV-2012-0208 | Ishier Raote Vivek Malhotra |
| Ministerio de Economía y Competitividad | BFU2013-44188-P | Ishier Raote Ishier Raote Vivek Malhotra |
| Ministerio de Economía y Competitividad | CSD2009-00016 | Ishier Raote Ishier Raote Vivek Malhotra |
| Ministerio de Economía y Competitividad | IJCI-2017-34751 | Ishier Raote Ishier Raote |
| National Institutes of Health | R35 GM118084 | James E Rothman |
| Ministerio de Economía y Competitividad | SEV-2015-0522 | Felix Campelo |
| Ministerio de Economía y Competitividad | RYC-2017-22227 | Felix Campelo |
| Fundació Privada Cellex | | Felix Campelo |
| Fundacio Privada Mir-Puig | | Felix Campelo |
| Generalitat de Catalunya | CERCA | Felix Campelo |

The funders had no role in study design, data collection and interpretation, or the decision to submit the work for publication.

### Author contributions

Ishier Raote, Frederic Pincet, Conceptualization, Resources, Data curation, Formal analysis, Supervision, Funding acquisition, Validation, Investigation, Visualization, Methodology, Writing - original draft, Project administration, Writing - review and editing; Andreas M Ernst, Conceptualization, Resources, Data curation, Formal analysis, Validation, Investigation, Visualization, Methodology, Writing - original draft, Writing - review and editing; Felix Campelo, Conceptualization, Data curation, Formal analysis, Funding acquisition, Validation, Investigation, Visualization, Methodology, Writing - original draft, Writing - review and editing; James E Rothman, Conceptualization, Supervision, Funding acquisition, Writing - original draft, Project administration, Writing - review and editing; Vivek Malhotra, Conceptualization, Formal analysis, Supervision, Funding acquisition, Methodology, Writing - original draft, Project administration, Writing - review and editing

### Author ORCIDs

Ishier Raote https://orcid.org/0000-0002-5898-4896
Felix Campelo http://orcid.org/0000-0002-0786-9548
Frederic Pincet https://orcid.org/0000-0002-4243-2157
Vivek Malhotra https://orcid.org/0000-0001-6198-7943

### Decision letter and Author response

Decision letter https://doi.org/10.7554/eLife.57822.sa1
Author response https://doi.org/10.7554/eLife.57822.sa2

## Additional files

### Supplementary files
• Transparent reporting form

### Data availability
All data generated or analysed during this study are included in the manuscript and supporting files.

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
