## [Decision Letter]

Thank you for submitting your article "TANGO1 membrane helices create a lipid diffusion barrier at curved membranes" for consideration by *eLife*. Your article has been reviewed by three peer reviewers, one of whom is a member of our Board of Reviewing Editors, and the evaluation has been overseen by Suzanne Pfeffer as the Senior Editor. The following individuals involved in review of your submission have agreed to reveal their identity: Benjamin S Glick (Reviewer #3); Jose Rizo-Rey (Reviewer #4).

The reviewers have discussed the reviews with one another and the Reviewing Editor has drafted this decision to help you prepare a revised submission.

The editors have judged that your manuscript is of interest, but as described below that additional experiments are required before it is published, we would like to draw your attention to changes in our revision policy that we have made in response to COVID-19 (https://elifesciences.org/articles/57162). First, because many researchers have temporarily lost access to the labs, we will give authors as much time as they need to submit revised manuscripts. We are also offering, if you choose, to post the manuscript to bioRxiv (if it is not already there) along with this decision letter and a formal designation that the manuscript is "in revision at *eLife*". Please let us know if you would like to pursue this option.

Summary:

In this "Research Advance" the authors build on their previous characterization of how the transmembrane protein TANGO1 resides at ER exit sites and mediates linkage to nearby ERGIC compartments. There are two key observations that together suggest a potentially novel and important role for TANGO1 in tubule-mediated transport- the prevention of intermixing of the membranes of donor and acceptor compartments. The first observation is that the TANGO1 membrane binding intramembrane helix called IM impedes lipid flux between a GUV and a tubule protruding from it. The second observation is that IM localizes TANGO1's cytoplasmic domain into clustered ring-like structures presumed to reflect targeting to the neck of budding ER export structures. There was agreement among the reviewers that these results will be of interest to the audience of e*Life*.

However, there were issues that need to be addressed. The first two will require additional experiments that should be straightforward, and it is agreed that they will significantly strengthen the conclusion that TANGO1 forms a membrane diffusion barrier that prevents the mixing of contents when ERGIC membranes fuse with the ER.

If performing one or both is not feasible because of the pandemic, the authors could try to address this with a textual revision, but it may be advisable to delay submission of a revised version until such experiments are possible again. There will be no deadline for completion.

1) How much of the IM-TM construct is present in the GUV relative to the lipids? Could it be so much that lipids fail to diffuse rapidly in the entire GUV? To show that a diffusion barrier exists specifically at the junction between the tube and the spherical portion, a suitable control would be to bleach half of the spherical portion of the GUV and confirm that fluorescence recovery within the spherical portion is unaffected by the IM-TM construct. This is a major concern that needs to be addressed, as it casts doubt on the main conclusion of the paper.

2) There is a need for a functional test. The authors analyze cells expressing Tango1∆IM-HA but do not report any tests for defects in compartmentalization. Their model predicts these cells should have loss of ERGIC-to-ER barrier function (e.g. ERGIC components redistributed to the ER among other changes).

3) The authors state that during procollagen export, the ER and ERGIC are transiently connected by a tunnel that keeps the two membranes distinct. Arguably, this an interesting hypothesis rather than an established fact. Such connections have not been directly visualized, and there is a diversity of views about how a large cargo traffics out of the ER. The language here should be modified to accurately reflect the current state of knowledge.

4) Please explain the rationale for choosing the single 6-minute time point for the fluorescence recovery assay. It seems like a long time for a lipid diffusion experiment.

5) Figure 2D is hard to process and lacks a statistical test. Because it's apparently important to the argument, a fuller explanation would help.

6) There is some uncertainty about the relevance of the experiments in Figures 3 and 4. This section could be strengthened with a clearer explanation of the questions that were being asked.

7) There is a concern regarding how the images of Figure 3—figure supplement 1 support the conclusion that TANGO1 and TANGO1ΔIM form small, discrete puncta, while TANGO1ΔTM does not.

8) The protein concentrations for the two constructs in the GUVs need to be shown equivalent after GUV formation (GFP intensity should suffice).

9) The conclusion "The two membrane helices together are not required to target TANGO1 to ERES" is confusing because the authors show the ∆TM construct is *not* localized to ERES. This construct contains the PRD and uses IM as a cryptic transmembrane domain and yet it is mislocalized. This seems to argue that both membrane domains are needed.

---

## [Author Response]

[…] There were issues that need to be addressed. The first two will require additional experiments that should be straightforward, and it is agreed that they will significantly strengthen the conclusion that TANGO1 forms a membrane diffusion barrier that prevents the mixing of contents when ERGIC membranes fuse with the ER.If performing one or both is not feasible because of the pandemic, the authors could try to address this with a textual revision, but it may be advisable to delay submission of a revised version until such experiments are possible again. There will be no deadline for completion.1) How much of the IM-TM construct is present in the GUV relative to the lipids? Could it be so much that lipids fail to diffuse rapidly in the entire GUV? To show that a diffusion barrier exists specifically at the junction between the tube and the spherical portion, a suitable control would be to bleach half of the spherical portion of the GUV and confirm that fluorescence recovery within the spherical portion is unaffected by the IM-TM construct. This is a major concern that needs to be addressed, as it casts doubt on the main conclusion of the paper.“How much of the IM-TM construct is present in the GUV relative to the lipids?” We have included a quantification of the protein content of the GUVs, relative to their lipid content. We show that the protein and lipid contents are similar across experimental treatments. We have also added an extensive description of this quantification, in the Materials and methods section.“… is fluorescence recovery within the spherical portion unaffected by the IM-TM construct?” To confirm that diffusion can occur freely, we quantify the rates of diffusion of IM-TM protein in the spherical portion of the GUV and show that the protein is predominantly monomeric and freely diffusing. These data have been added as Figure 2—figure supplement 1) with a detailed description of our approach, in the Materials and methods section.2) There is a need for a functional test. The authors analyze cells expressing Tango1∆IM-HA but do not report any tests for defects in compartmentalization. Their model predicts these cells should have loss of ERGIC-to-ER barrier function (e.g. ERGIC components redistributed to the ER among other changes).

A functional demonstration of the diffusion barrier in vivo would be wonderful. Unfortunately, we do not think it can be achieved at this stage.

Since TANGO1 recruits ERGIC membranes to the ER exit site, the mixing of ER and ERGIC material will itself depend on TANGO1. We have previously shown that depleting TANGO1 partially abrogates ERGIC recruitment to ER exit sites at collagen accumulations. It is unclear what effect the IM helix will have on TANGO1 retrograde recruitment of ERGIC membrane.

in vivo, an ER-ERGIC barrier will be affected by a number of components, including other TANGO1-family proteins and their isoforms, all of which are still present in cells after full-length TANGO1 is depleted. There are several possible explanations for any changes in the distribution of ER and ERGIC components under our experimental conditions, identifying each and their relative contributions to a barrier are not feasible, making it virtually impossible to interpret any observations.

Our data present a novel insight into the existence of a barrier, delineating the mechanisms involved in vivo will be an exciting avenue of research, but beyond the scope of this study.

3) The authors state that during procollagen export, the ER and ERGIC are transiently connected by a tunnel that keeps the two membranes distinct. Arguably, this an interesting hypothesis rather than an established fact. Such connections have not been directly visualized, and there is a diversity of views about how a large cargo traffics out of the ER. The language here should be modified to accurately reflect the current state of knowledge.

We have included a paragraph in the Introduction, describing alternate models of procollagen export.

4) Please explain the rationale for choosing the single 6-minute time point for the fluorescence recovery assay. It seems like a long time for a lipid diffusion experiment.

We used a single data point at 6 minutes because we wanted to be close to the fluorescence recovery plateau and we also wished to avoid spontaneous bleaching to have measurement as accurate as possible. We have stated this explicitly in the text and included detailed reasoning in the Materials and methods section.

5) Figure 2D is hard to process and lacks a statistical test. Because it's apparently important to the argument, a fuller explanation would help.

To convey the information better, we have replaced the cumulative frequency plot with a frequency histogram. We now include a paragraph to describe this plot.

6) There is some uncertainty about the relevance of the experiments in Figures 3 and 4. This section could be strengthened with a clearer explanation of the questions that were being asked.

We presume the reviewers are referring to Figure 3 and Figure 3—figure supplement 1. In this experiment we test one potential way that TANGO1 could utilize the membrane shape-sensing module of two adjacent membrane helices. A first possibility we could test is that it promotes the correct localisation of TANGO1 to an ERES, as a site of appropriately shaped membrane. We show that deleting the IM helix has little effect on targeting TANGO1 to an ERES, discard this proposal, and move to another possibility in the next section. We have altered the text to make this explanation clearer.

7) There is a concern regarding how the images of Figure 3—figure supplement 1 support the conclusion that TANGO1 and TANGO1ΔIM form small, discrete puncta, while TANGO1ΔTM does not.

The key observations we made in this section are that TANGO1 proteins lacking either the IM or TM helices behave slightly differently, but both are still targeted to ERES and collagen accumulations. We have changed the text to clarify that these are controls to show the TANGO1 proteins we express retain some of their functions and are not entirely dysfunctional

8) The protein concentrations for the two constructs in the GUVs need to be shown equivalent after GUV formation (GFP intensity should suffice).

As described above (point 1), we quantified GFP expression in the GUVs and show that the different conditions have similar levels of protein incorporated into GUVs.

9) The conclusion "The two membrane helices together are not required to target TANGO1 to ERES" is confusing because the authors show the ∆TM construct is not localized to ERES. This construct contains the PRD and uses IM as a cryptic transmembrane domain and yet it is mislocalized. This seems to argue that both membrane domains are needed.

See point 6 above, we have changed the text and title of the section to clarify our statements and avoid confusion. The model proposed is that the two helices act as a membrane-shape sensor of sorts. In this section, we asked whether the shape-sensitivity (provided by the two helices *together*) contributed to targeting TANGO1 to an appropriate site, such as an ERES, based on membrane shape/curvature. Deleting the IM helix would abrogate the membrane-shape sensitivity, yet it had no discernible effect on targeting TANGO1 to an ERES. Therefore, we conclude the membrane-shape sensor is not a major determinant of TANGO1 localisation to an ERES.